# The dark exciton ground state promotes photon-pair emission in individual perovskite nanocrystals

Philippe Tamarat[1,2,8], Lei Hou [1,2,8], Jean-Baptiste Trebbia[1,2], Abhishek Swarnkar[3,4], Louis Biadala [5], Yann Louyer[6], Maryna I. Bodnarchuk[3], Maksym V. Kovalenko [3,4], Jacky Even [7] & Brahim Lounis [1,2✉]

Cesium lead halide perovskites exhibit outstanding optical and electronic properties for a wide range of applications in optoelectronics and for light-emitting devices. Yet, the physics of the band-edge exciton, whose recombination is at the origin of the photoluminescence, is not elucidated. Here, we unveil the exciton fine structure of individual cesium lead iodide perovskite nanocrystals and demonstrate that it is governed by the electron-hole exchange interaction and nanocrystal shape anisotropy. The lowest-energy exciton state is a long-lived dark singlet state, which promotes the creation of biexcitons at low temperatures and thus correlated photon pairs. These bright quantum emitters in the near-infrared have a photon statistics that can readily be tuned from bunching to antibunching, using magnetic or thermal coupling between dark and bright exciton sublevels.

[1] Université de Bordeaux, LP2N, F-33405 Talence, France. [2] Institut d'Optique and CNRS, LP2N, F-33405 Talence, France. [3] Empa-Swiss Federal Laboratories for Materials Science and Technology, CH-8600 Dübendorf, Switzerland. [4] Department of Chemistry and Applied Biosciences, Institute of Inorganic Chemistry, ETH Zürich, CH-8093 Zürich, Switzerland. [5] Institut d'Electronique, de Microélectronique et de Nanotechnologie, CNRS, Villeneuve-d'Ascq, France. [6] LOMA, CNRS UMR 5798, University of Bordeaux, F-33400 Talence, France. [7] Univ Rennes, INSA Rennes, CNRS, Institut FOTON - UMR 6082, F-35000 Rennes, France. [8] These authors contributed equally: Philippe Tamarat, Lei Hou. ✉email: brahim.lounis@u-bordeaux.fr

Recent advances in the colloidal synthesis of strongly emitting perovskite nanocrystals (NCs) open up new opportunities for the fabrication of tunable light sources based on the composition and quantum size effect tuning[1], such as light-emitting diodes and lasers[2], and for the exploration of their potential use as quantum light sources[3–6]. The knowledge of the band-edge exciton fine structure in perovskites and the ability to tailor it with NC composition, morphology, or external fields are of prime importance for the development of efficient single-photon sources or sources of entangled photons for quantum information processes[7–9], and for applications in optoelectronics and spin-based technologies[10–12]. Yet, the bright or dark character of the ground exciton in perovskites is the subject of continuing debate. The band-edge exciton of perovskites is formed with a hole (h) in an S-like state with angular momentum $J_h = 1/2$ and an electron (e) in a spin-orbit split-off state with angular momentum $J_e = 1/2$[13,14]. The resulting four-fold degenerate exciton level is split by the electron–hole exchange interaction into a dark ground singlet state ($J = 0$) and a bright triplet ($J = 1$). At liquid helium temperatures, single cesium lead halide ($CsPbX_3$) perovskite NCs display a bright photoluminescence (PL) with a spectral fingerprint assigned to the bright triplet structure[15–20]. It has been speculated that this brightness takes its origin in a polar lattice symmetry breaking that would reverse the exciton sub-levels ordering by a Rashba effect and thus place the bright triplet exciton below the dark singlet[20,21]. While the spectral signature of a dark exciton lying below the bright triplet has been found in hybrid organic–inorganic formamidinium lead bromide ($FAPbBr_3$) NCs[22], theoretical calculations still predict order inversion in $CsPbX_3$ NCs[23] and thus far, no signature of a low-lying dark exciton state has been found in such NCs.

Here, we demonstrate that the band-edge exciton fine structure of inorganic $CsPbI_3$ NCs presents a dark ground singlet state. We fully explain the fine structure splittings with a theoretical model solely based on the electron–hole exchange interaction and NC shape anisotropy. In the view of applications to quantum technologies, we introduce resonant PL excitation of the excitonic sublevels to investigate the indistinguishability character of the photons emitted by these NCs. Importantly, we demonstrate that the presence of a long-lived ground exciton state favors the formation of biexcitons and thus the emission of pairs of correlated photons. We show that this is a general behavior that can be observed for perovskite NCs as well as for conventional CdSe quantum dots. This property makes single $CsPbI_3$ NCs versatile bright, quantum light sources in the near infrared with photon statistics that can be tuned from bunching to antibunching, using magnetic coupling or thermal mixing between dark and bright exciton states.

## Results

**Spectral signatures of the NCs' luminescence.** The $CsPbI_3$ NCs of this study are synthesized with an orthorhombic γ-phase crystal structure[24,25] (Fig. 1a). They display cuboid shapes with an average side of $11.2 \pm 1.2$ nm and an apparent aspect ratio of 1.13 on average from electron microscopy imaging (see Fig. 1b, c). When dissolved in hexane, NCs exhibit a bright red PL centered at 1.81 eV, with a high color purity and a PL quantum yield exceeding 65% at room temperature. The PL peak energy is slightly blue-shifted as compared with the bulk material (~1.73 eV)[26], indicating a regime of weak quantum confinement in these NCs with a mean size slightly larger than twice the exciton Bohr radius $a_B = 4.6$ nm in $CsPbI_3$[27]. Using a home-built scanning confocal microscope operating at cryogenic temperatures, we have investigated the PL spectra of single NCs dispersed in a polystyrene matrix and weakly

excited at a laser wavelength of 561 nm (see "Methods" section for details). Their emission energy centered on 1.77 eV (Fig. 1a) is red-shifted with respect to the room temperature value, due to a displacement of the valence band energy with lattice contraction[28]. At low temperature (~4 K), thermal dephasing is reduced and the PL spectra of single NCs present sharp peaks attributed to the zero-phonon emission lines (ZPLs) resulting from the recombination of the charge complexes formed in the NC, as exemplified in the typical spectral trail displayed in Fig. 1d. This trail shows an emission switch of a NC from its neutral state, displaying ZPLs labeled X for the exciton and XX for the biexciton[16], to its charged state characterized by a single ZPL labeled X*. As shown in Fig. 1e, integrating the spectra gives greater details of the fine structure of these charge complexes, as well as the longitudinal optical (LO) phonon replicas of the ZPLs. The exciton fine structure is revealed both through the biexciton-to-exciton transitions and through the bright exciton recombination. Mapping the biexcitonic transition multiplet XX onto the excitonic one X (see inset of Fig.1e) shows evidence for a perfect match in their spectral structures and allows a straightforward correspondence of the transition lines (see also Supplementary Fig. 1). As exemplified in Fig. 2 and Supplementary Fig. 2, this spectroscopic study reveals one-line, two-line, or three-line PL spectra in proportions of ~5%, 30%, and ~65%, respectively. Within a resolution of ~120 μeV, these spectral structures result from a subtle interplay of crystal structure and shape anisotropy effects[15,29,30]. For instance, assuming a cubic symmetry of the NC crystal structure (resp. shape), shape (resp. crystal structure) anisotropy governs the number of bright levels: Single-line, two-line, and three-line spectra will be, respectively, assigned to cubic, tetragonal, and orthorhombic anisotropy of the NC shape (resp. crystal structure).

**Exciton fine structure and signature of the dark singlet.** The spectral signature of the dark singlet exciton is evidenced under the application of an external magnetic field, as demonstrated in Fig. 2 (see other examples in Supplementary Fig. 2). Regardless of the triplet structure, a new emission line indeed emerges from the LO-phonon sidebands, a few meV below the bright triplet, as a signature of magnetic brightening of a low-lying dark exciton state in these inorganic lead halide perovskites. Despite this level ordering, perovskite NCs display an intense luminescence at low temperature because of an extremely reduced bright-to-dark phonon-assisted relaxation[22,31], a remarkable property of these materials as compared to more conventional III–V or II–VI semiconductors. Indeed, spin relaxation processes involving polar acoustic phonons related to piezoelectricity are absent[32], and carrier–phonon interactions via deformation potentials are reduced in these structurally soft semiconductors with strong lattice anharmonicity affecting the optical modes[33]. The structure of the optical phonon modes of the $CsPbI_3$ NCs is detailed in Supplementary Fig. 3 and compared with that of $CsPbBr_3$ NCs. It shows evidence that the bright-to-dark relaxation remains inhibited even when the bright-dark splitting coincides with the phonon modes. The fact that phonons do not carry angular momentum likely explains why the bright-to-dark relaxation is inhibited in perovskites[34]. Indeed, it requires Raman-like phonon processes involving two-phonon modes whose energy difference matches the bright-dark splitting[22,31] (see Fig. 2d and Supplementary Fig. 4). Interestingly, a direct signature of extremely slow phonon relaxation within the bright sublevels is the equality in relative weights between the corresponding XX and X triplet components (see Fig. 1e and Supplementary Fig. 1). It proves that the relaxation is slower than the exciton recombination, whose lifetime is on the nanosecond time scale. This points to a

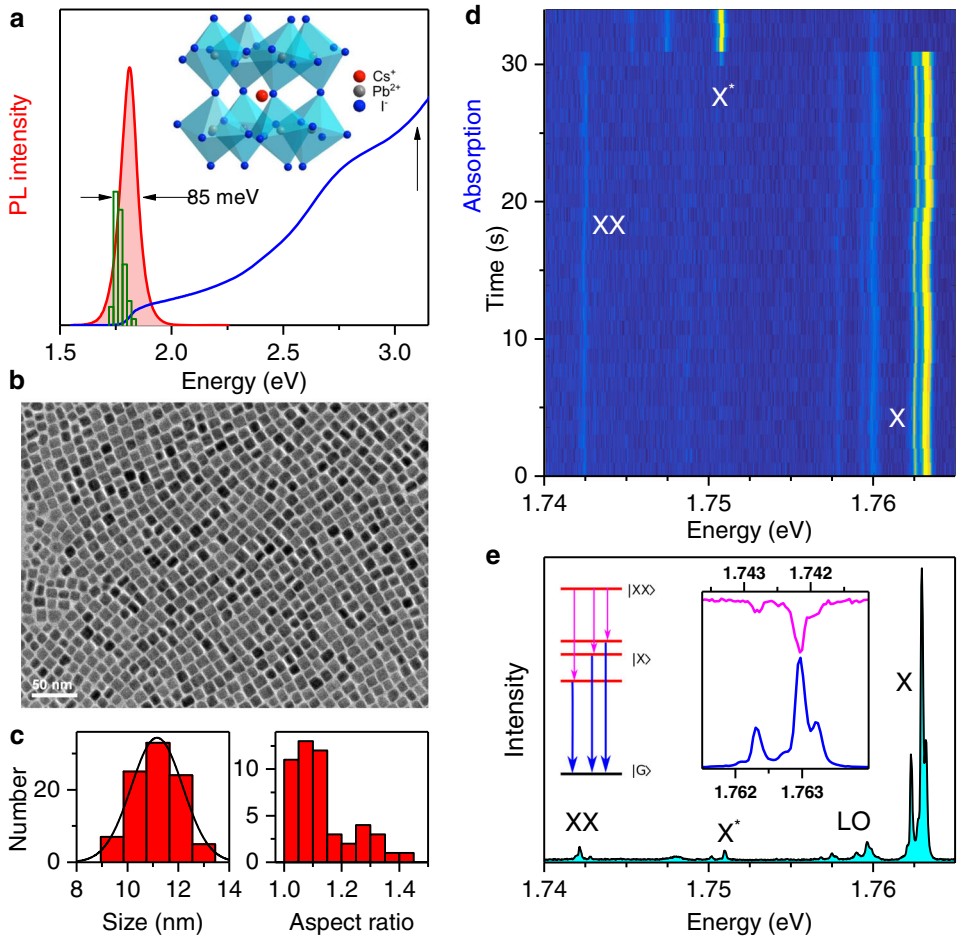

**Fig. 1 Structural and optical characteristics of CsPbI₃ perovskite NCs. a** Orthorhombic perovskite crystal structure of CsPbI₃ NCs. Room-temperature absorption and PL spectra of an ensemble of CsPbI₃ NCs dispersed in hexane. The vertical arrow at 3.1 eV indicates the excitation energy chosen for the acquisition of this PL spectrum. The width (FWHM) of the PL spectrum is ~85 meV. The green histogram displays the distribution of emission energies of 58 single NCs at 4 K. **b** Annular bright-field scanning transmission electron microscopy image of CsPbI₃ NCs. **c** Size histogram (left panel) and histogram of aspect ratios (right panel) for these NCs. The black curve is a Gaussian distribution centered at 11.2 nm, with a full-width at half-maximum (FWHM) of 2.4 nm. The mean aspect ratio is 1.13. **d** Spectral trajectory of a single NC at 4 K, built with 33 consecutive PL spectra recorded over 1 s at an excitation intensity of ~50 W cm⁻². It shows a spectral jump together with a change of spectral structure associated to a transition between exciton emission and trion (charged exciton) emission. **e** PL spectrum of the same single NC at 4 K, integrated over 57 spectra of 1 s extracted from **d**. It displays the spectral fingerprints of the exciton (X) band-edge exciton fine structure with its LO-phonon replica, trion (X*) and biexciton (XX) emissions. The correspondence between the XX and X multiplets is shown in the inset. As depicted in the energy level diagram, the transition lines connect the bright biexciton state |XX> with zero angular momentum to the non-degenerate exciton triplet |X>, and the latter to the NC ground state |G>.

negligible Rashba-like spin–orbit coupling[35–37] that would allow one-phonon bright-to-dark relaxation in these materials.

We have studied the exciton fine structure of tens of individual NCs and find a clear increase of the bright triplet splittings (up to ~1 meV) and the bright-dark splittings (up to ~5 meV) with the exciton recombination energy, as shown in Fig. 3a, b and Supplementary Fig. 5. The trend of such a correlation also shows up for the bright triplet splittings of CsPbBr₃ NCs (see Supplementary Fig. 6) and CsPbBr₂Cl NCs[20]. These results demonstrate that fine structure splittings are dominated by the exchange interaction under the combined effects of quantum confinement and screening of the electron–hole interaction. Indeed, the contribution of the Rashba effect to the bright state splittings has been predicted to decrease with increasing NC sizes[38], which points to a reduced contribution of this effect in cesium lead halide perovskite NCs. In order to reproduce these results, we have calculated the exchange interaction for cuboid shaped NCs with a cubic crystal phase in the weak quantum confinement regime, using

a variational approach based on an effective mass model for the monoelectronic states (Supplementary Note 1). Figure 3c shows the results obtained in the case of a tetragonal NC shape, where the bright sublevels form a doublet comprising a non-degenerate sublevel (Z state) defined by the NC elongation axis and a two-fold degenerate sublevel (X and Y states). Their splitting increases with the NC aspect ratio (see also Supplementary Fig. 7), and their ordering becomes reversed when the NC shape switches from flattened to elongated. This is illustrated in Fig. 3d, where the PL spectra of two nearly tetragonal NCs with different aspect ratios are presented (NCs with sides $L_x \approx L_y > L_z$ for the left spectrum, and $L_x \approx L_y < L_z$ for the right spectrum). Importantly, our observations are in contradiction with the assumption of a Rashba effect that would always place the bright Z level below the twofold degenerate X–Y level[23]. This is also the case in CsPbBr₃ NCs, as shown in Supplementary Fig. 6. Finally, we find that the evolution of the calculated bright-dark splitting as a function of the exciton

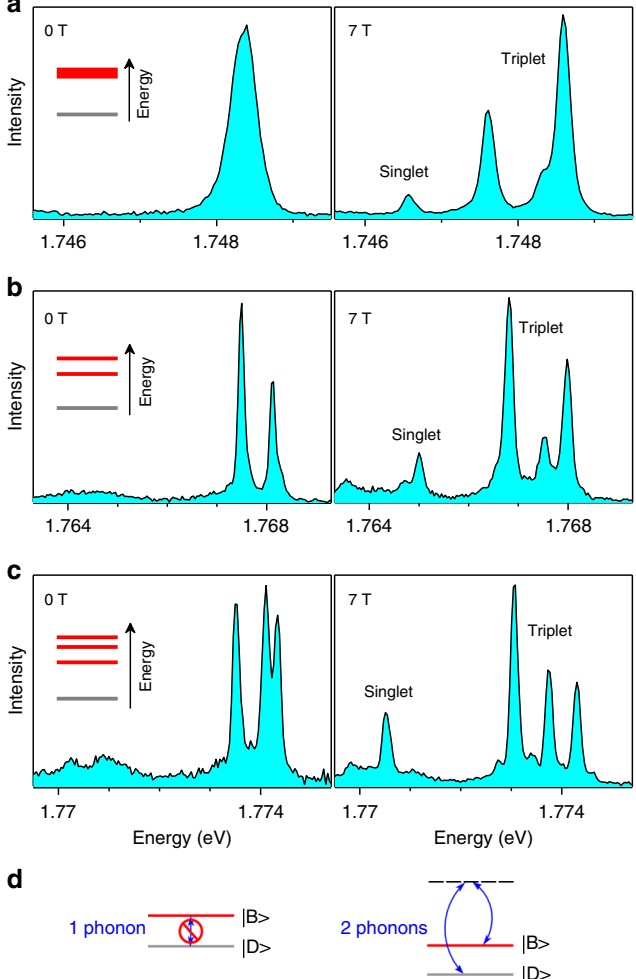

**Fig. 2 Magnetic brightening of the ground dark singlet exciton of single CsPbI₃ NCs. a–c** Spectra of three different single NCs at 4 K, which present one **a**, two **b**, three **c** exciton recombination ZPLs in zero field. The insets show the corresponding diagrams of exciton fine structure, which are sensitive to shape anisotropy. The levels displayed in red represent the optically active triplet states, while the ground level in gray is the singlet dark state. At 7 T, magnetic splitting and coupling among the fine structure sublevels reveal the entire spectral fingerprint of the bright triplet and the lowest-energy singlet state. **d** Bright-to-dark relaxation with a one-phonon process is inhibited in lead halide perovskites (left panel). Thermal mixing between dark and bright states occurs via a two-phonon process (right panel).

recombination energy (Fig. 3e) is in quantitative agreement with the experimental results when taking into account the effect of dielectric confinement on the exchange interaction. More quantitative comparison between theory and experiment, in particular the assignment of non-degenerate bright triplet ZPLs to the levels X, Y, and Z, would require refined models that take into account crystal phase and shape anisotropies[39] together with the knowledge of the low-temperature crystal structure, 3D shape, orientation, and dielectric environment of each NC.

In order to predict the offset of the dark ground state in the fine structure of lead halide perovskites, we have calculated the exchange interaction in bulk perovskites with various compositions and plotted them as a function of their exciton Bohr radius in Fig. 3f. The weakest value is obtained for FAPbI₃ perovskites. It is consistent with spectroscopic studies of weakly confined single FAPbI₃ NCs, which present a narrow exciton fine structure within a few

hundred μeV that could not be resolved under a magnetic field of 7 T[31]. CsPbBr₃ perovskites display the largest bulk exchange interaction, explaining why the distributions of bright triplet splittings measured in CsPbBr₃ NCs[15,18,20] are larger than those of FAPbBr₃ and CsPbI₃ NCs. Despite spectroscopic investigations on different CsPbBr₃ NCs with different sizes, shapes, and surface ligands, we could not observe magnetic brightening of the dark state with magnetic fields up to 7 T[15] for different exciton confinement regimes (NCs with sizes ~4 to ~20 nm) and different surface ligands. Since magnetic brightening is inversely proportional to the square of the bright-dark splitting, it is likely negligible at such fields due to a large offset between the dark ground singlet state and the bright triplet in these NCs. This picture is supported by the onset of a long-time component in the PL decay of ensembles of CsPbBr₃ NCs at fields higher than 10 T[40], as an indirect signature of a low-lying long-lived sublevel[41].

**Exciton optical coherence lifetime.** Due to their bright PL in the near-infrared and sharp ZPLs at low temperature, single CsPbI₃ NCs are attractive solid-state emitters for a potential use as quantum light sources. In particular, sources that deliver single indistinguishable photons on demand are required for a number of quantum information processing schemes[7]. Such photons stem from the ZPL when its spectral linewidth $1/\pi T_2$ reaches its fundamental lower bound $1/(2\pi T_1)$, where $T_2$ is the optical coherence lifetime of the emission and $T_1$ the emitting state lifetime. Time-resolved photon-correlation Fourier spectroscopy has provided a coherence lifetime $T_2 \sim 80$ ps for individual CsPbBr₃ NCs[6]. However, time-resolved approaches without spectral filtering fail at identifying the charge complexes at the origin of the emission. For CsPbI₃ NCs, a similar coherence lifetime has been extracted from first-order PL correlation measurements[42]. In these early investigations, the excess energy between absorption and emitted photons may induce spectral diffusion of the transition energy[31]. We thus measure the ultimate limit of the optical coherence lifetime of perovskite NCs through resonant excitation on their excitonic ZPLs. Resonant excitation spectra are recorded by scanning a single-mode laser across one of the fine structure ZPLs, while collecting the fraction of PL emitted on the LO phonon replicas, as shown in Fig. 4a. A narrow Lorentzian ZPL is presented in Fig. 4b with a homogeneous linewidth as narrow as ~5 GHz FWHM, corresponding to an optical coherence time $T_2 \sim 64$ ps. This value is still short in comparison with the exciton lifetime $T_1 \sim 1$ ns measured on the same NC (see Fig. 4c). A possible origin of ZPL broadening is fast-spectral diffusion induced by fluctuations of the local electric field and dielectric screening, as a result of charge noise produced by charge displacements at the surface of the NCs or in its local environment. These processes are likely the same as those responsible for slow spectral diffusion accompanied by variations of the exciton fine-structure splittings. As shown in Supplementary Fig. 8, the fine-structure splittings increase with the exciton recombination energy, which suggests that the spectral jumps take their origin in fluctuations of the dielectric confinement that affects both the transition energy and the electron–hole exchange interaction. Despite these homogeneous broadening mechanisms, our high-resolution PL excitation method provides a precious tool to investigate subtle lifts of degeneracy in the exciton fine structure, that would be buried within the spectral resolution of the PL spectra. In further investigations, such variations of the exciton fine structure should be correlated with the 3D characterization of the NCs morphology.

**Tuning the photon statistics of single NCs.** We now study the statistics of the photons emitted by individual CsPbI₃ NCs, using

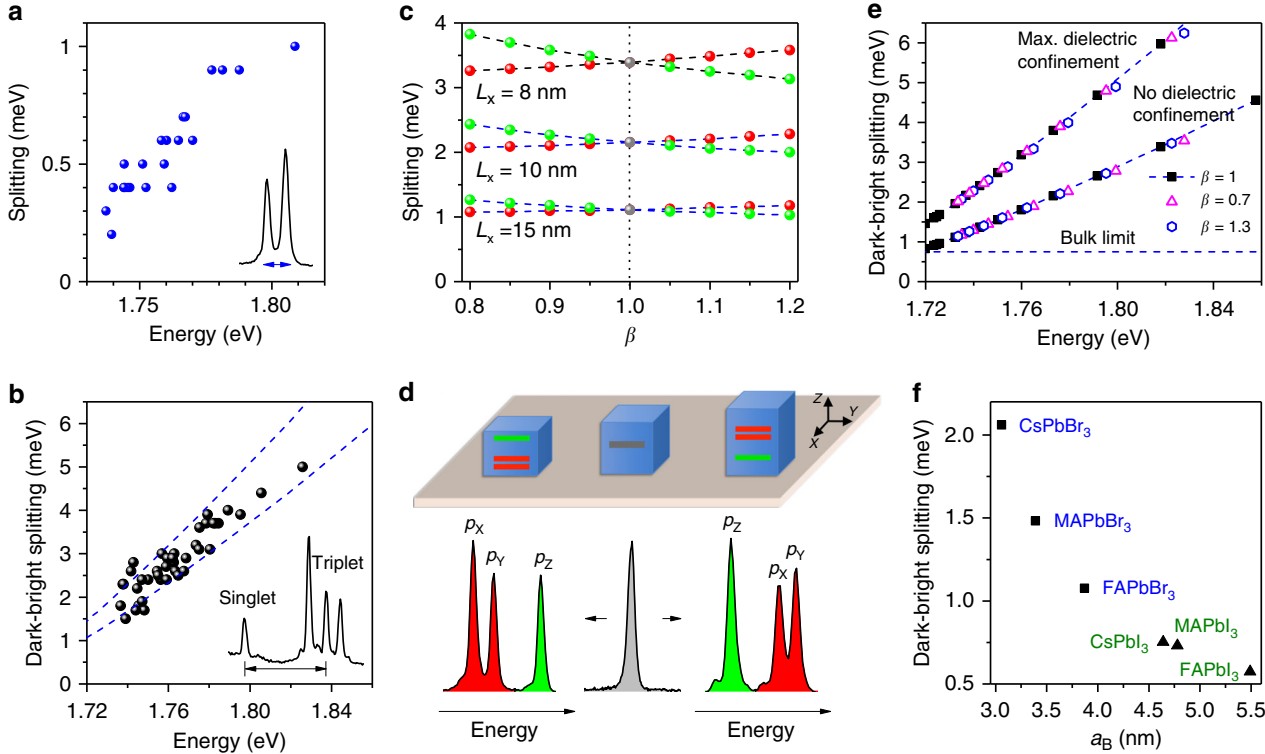

**Fig. 3 Effects of quantum and dielectric confinements on the exciton fine structure. a** Statistics of zero-field splittings for the two-line spectra as a function of the lowest emission line energy. **b** Splitting at 7 T between the singlet line and the central triplet line, plotted for 40 NCs as a function of the exciton recombination energy. The dashed lines are simulations of the electron–hole exchange interaction of cubic-shaped NCs, considering the extreme values of the dielectric constant of the NC local environment: Vacuum (top line) and sapphire (bottom line), see Supplementary Note 1. **c** Computed evolution of the triplet splitting with the aspect ratio $\beta$ of a tetragonal-shaped NC with a cubic crystal structure. The cube is elongated along the $Z$ axis, with an aspect ratio defined as $L_x = L_y = L_z/\beta$. The origin of the energy scale is taken at the ground singlet level. The splittings are presented for three different NC sizes (covering the range of the explored NC sizes) and increase with quantum confinement. The effects of dielectric confinement, which enhance the fine structure splittings, are not taken into account in **c**. **d** Three PL spectra are chosen to illustrate the assignment of the triplet spectral structure to the NC morphology (slightly distorted tetragonal shape), assuming a cubic lattice. **e** Computed evolution of the exchange interaction of a tetragonal-shaped NC as a function of the exciton recombination energy, without dielectric confinement (lower branch) and with the maximal correction factor for the dielectric confinement effect, obtained when the external medium of the NC is considered as vacuum (upper branch). Data points are presented for several average NC sizes and three aspect ratios: $\beta = 0.7$ (triangles), 1 (squares), 1.3 (hexagons), in order to show the influence of the NC aspect ratio on the exciton recombination energy and fine structure splittings. **f** Exchange interaction of several bulk lead halide perovskites with different chemical compositions, plotted as a function of their exciton Bohr radius.

a Hanbury Brown and Twiss coincidence setup to measure the histograms of time delays between consecutive photon pairs. A striking feature is the strong bunching observed at low temperatures with the PL signal formed by all emission lines, as exemplified in Fig. 5a. We attribute this behavior to the exciton-shelving role played by the long-lived ground exciton state, which favors the generation of biexcitons in the NCs, followed by biexciton to bright-exciton-correlated photon pairs. The photon bunching character weakens when the temperature is raised. This results from thermal mixing between bright and dark states, which shortens the effective lifetime of the dark state. As this lifetime approaches its limit given by twice the bright exciton lifetime, photon bunching turns to antibunching (see Fig. 5a and Supplementary Figs. 4, 9) with a zero-delay autocorrelation function set by the biexciton radiative yield. This behavior is well reproduced with simulated autocorrelation functions (Fig. 5b) deduced from the solutions of rate equations in a simple four-level model including the zero-exciton ground level, thermally mixed bright and dark exciton sublevels and a biexciton level (see Supplementary Note 2). To further support the role of the dark state on the photon statistics, we have studied the evolution of the autocorrelation function with magnetic fields. Clearly,

magnetic coupling of dark and bright states leads to shortening of the long component of the PL decay and to weakening of the photon bunching, as shown in Fig. 5c (see other examples in Supplementary Figs. 9–11). It is worth noting the systematic evolution of the photon statistics from bunching to antibunching when the excitation intensity is increased (Supplementary Fig. 12). We attribute this dependence to the growing depopulation of the dark exciton state through the formation of a biexciton, whose radiative recombination solely populates the bright exciton states.

In order to generalize the picture of a long-lived ground state promoting photon pairs in quantum dots, we have performed similar experiments on CdSe core–shell NCs (see Supplementary Fig. 13), which present a ground "dark" exciton sublevel located a few meV below the lowest bright state[43]. Strong photon bunching is observed for single CdSe NCs at liquid helium temperatures and weak intensities. However, the transition from bunching to antibunching occurs at much lower temperatures than for perovskites, as soon as the thermal energy approaches the dark-bright splitting. Indeed, thermalization between fine structure sublevels is mediated in these NCs by one acoustic phonon whose energy matches the bright-dark splitting[44,45], while in perovskite

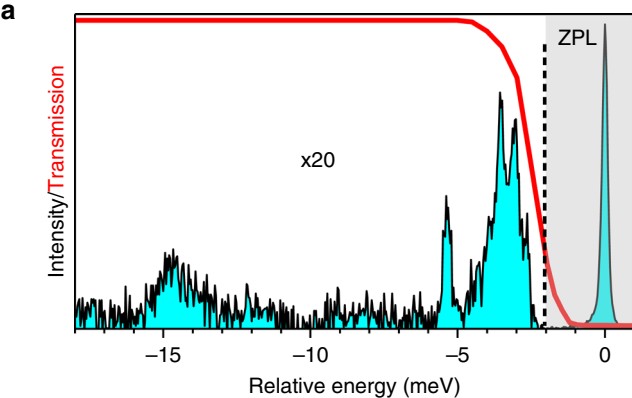

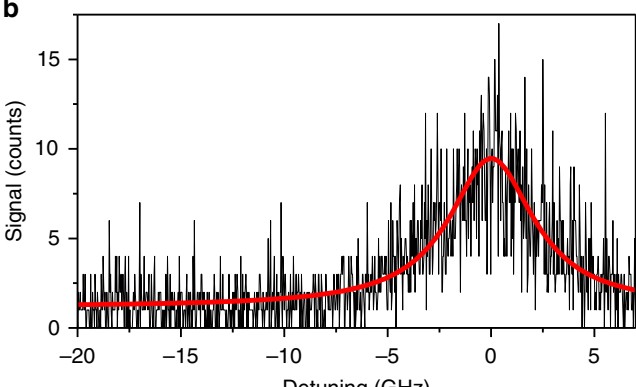

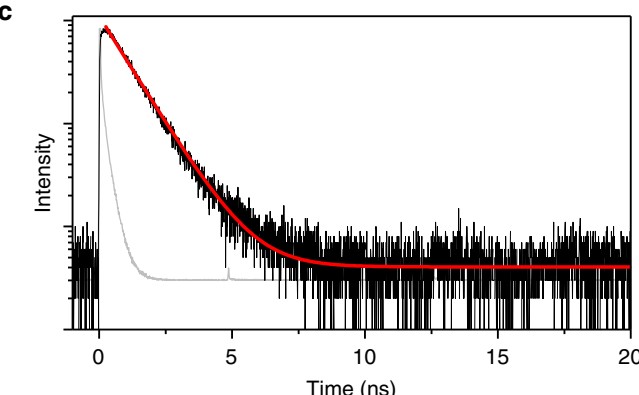

**Fig. 4 Measurement of the exciton optical coherence lifetime. a** Low-temperature PL spectrum of a NC presenting a single ZPL. The origin of the energy scale is taken at the ZPL. The low-energy part of the spectrum is magnified by a factor of 20. Six LO phonon modes are identified with phonon energies 3.0, 3.5, 5.4, ~8, ~11, ~15 meV. To record a resonant PL excitation spectrum, the frequency of a single-mode laser is scanned across a ZPL, while photons emitted on the highest phonon-energy LO phonon sidebands are collected through a low-pass filter (transmission curve illustrated by the red line). **b** Resonant PL spectrum built from a sum of 9 scans (1 s per scan, 1 ms per bin), with an excitation intensity of 20 W cm$^{-2}$. It is fitted with a Lorentzian profile (red curve) with a FWHM linewidth of 5 GHz. **c** Typical PL decay of a single NC, for an average number of exciton per pulse $\langle N \rangle = 0.2$. The red curve is an exponential fit with a lifetime of 1.0 ns. The instrument response function is shown in gray.

NCs it occurs through a second-order process with high-energy phonons.

Finally, cross-correlation measurements complement the identification of the biexciton emission in perovskite NCs and show their suitability as sources of correlated photon pairs. As shown in Fig. 5d,

energy-resolved photon cross-correlation histograms display an asymmetric bunching for positive delay times, owing to the cascaded emission of photon pairs with a biexciton recombination photon preceding an exciton recombination photon. Strong photon antibunching at low temperature is retrieved in single CsPbI$_3$ NCs when filtering the emission from the exciton recombination lines.

## Discussion

This study addresses the debate concerning the physical interactions at the origin of the band-edge exciton fine structure in lead halide perovskites. In particular, the singlet–triplet sublevel ordering and the triplet splittings are set by the electron–hole exchange interaction, which is exalted by quantum confinement and affected by the NC shape anisotropy. The remarkable brightness of these nanoemitters at low temperature takes its origin in the efficient inhibition of the phonon-assisted relaxation between the band-edge exciton sublevels. Moreover, the low-lying long-lived ground exciton state can be exploited to tune the quantum properties of the light emitted by perovskite NCs. Compared to CdSe NCs that have benefited from decades of developments and now achieve near-unity PL quantum yields[46], leading to the realization of CW lasing with these materials[47], synthesis of perovskite NCs is still in its infancy. One can foresee important developments motivated by the promising prospects of these materials as classical or quantum light sources. The next investigations will aim at reducing the dephasing rate and spectral diffusion in these materials and improve the indistinguishability character of the emitted photons, with the application of strong local electric fields that stabilize the charge distributions in the NC, or further surface passivation with suitable ligands or the growth of inorganic shells. In general, photon pairs emitted by conventional quantum dots undergo various entanglement degrading effects such as band mixing, exciton spin-flip processes and fine structure splitting, which drastically reduce their degree of entanglement. With a view to achieving ideal sources of entangled photons, perovskite NCs exhibit pure electron and hole bands, extremely slow spin-flip relaxation and, for a fraction of them, degenerate bright triplet emission.

## Methods

**Preparation of cesium-oleate precursor.** In a 100 mL three-necked round bottom flask a mixture of 0.5 g Cs$_2$CO$_3$, 2 mL OA, and 50 mL ODE is dried at 120 °C for 2 h under vacuum, followed by purging with nitrogen for 30 min. The reaction is finished when the Cs$_2$CO$_3$ powder dissolves completely to give a clear solution. The prepared solution of cesium oleate in ODE (0.06 M) is used as cesium precursor and stored under nitrogen atmosphere.

**Synthesis of colloidal CsPbI$_3$ NCs.** In a 100 mL three-necked round bottom flask, 200 mg PbI$_2$ (0.43 mmol) are suspended in ODE (10 mL). The reaction mixture is degassed at 100 °C under vacuum for 3 h. Subsequently, 1 mL OA, 1 mL OAm, and 1 mL TOP are added under nitrogen atmosphere. The reaction temperature is increased to 175 °C and preheated (~70 °C) Cs-oleate solution (1.6 mL, 0.06 M) is swiftly injected to the reaction flask. In 10 s the reaction mixture is cooled by a water-ice bath.

**Purification of CsPbI$_3$ NCs.** Purification of the synthesized NCs is conducted in a glovebox under inert atmosphere. In order to remove the excess of ligands, 35 mL anhydrous MeOAc are added to the crude solution, then centrifuged at 12,100 rpm for 1 min (20,130 × g, centrifuge: Eppendorf 5810R). The supernatant is discarded and the precipitate is dissolved in ~5 mL anhydrous hexane. The resulting solution is centrifuged at 4000 rpm for 10 min and the precipitate is discarded. The obtained solution of CsPbI$_3$ NCs (supernatant) is used for further studies.

**Single NC spectroscopy.** The NCs in hexane are kept under Ar atmosphere for further analysis. For measurements on single NCs at cryogenic temperatures, a dilute solution of CsPbI$_3$ NCs is mixed with a 2 wt% polystyrene solution and spin-coated onto a clean sapphire coverslip with a rotation speed of 2000 rpm. A home-built scanning confocal microscope based on a 0.95 numerical aperture objective is placed in a cryostat and used to image single NCs by raster scanning the sample. For the acquisition of PL spectra, the NCs are excited at 561 nm with a cw laser

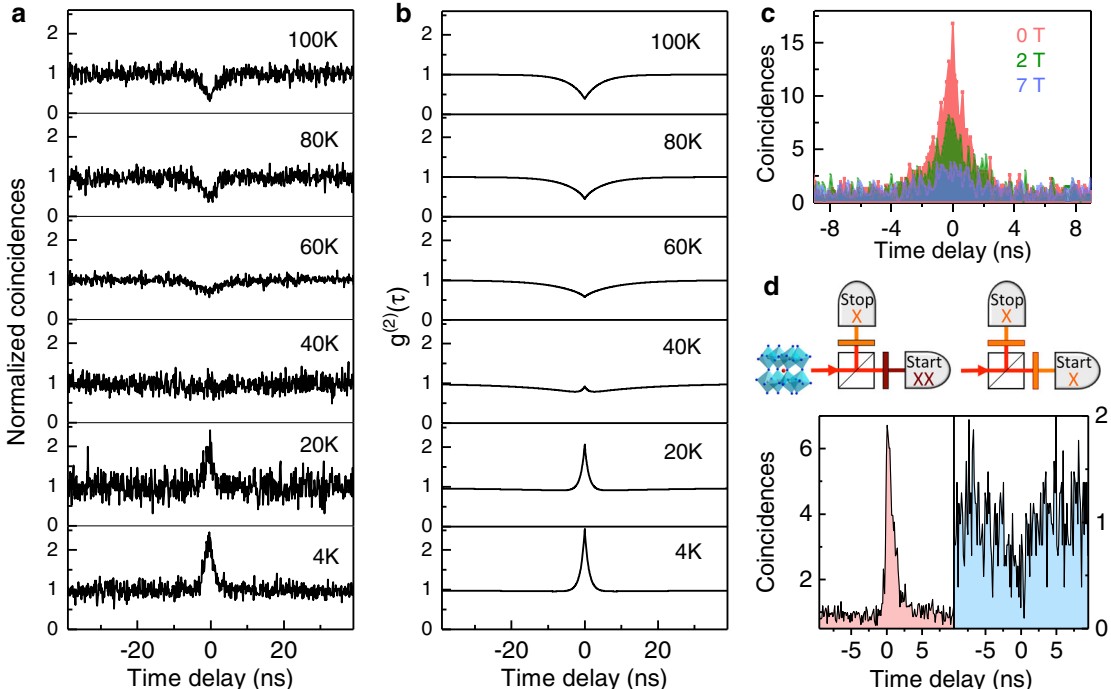

**Fig. 5 Tuning the photon statistics of single NCs with temperature and magnetic field. a** Normalized photon coincidence histograms of a NC for various temperatures, in zero field and at ~100 W cm$^{-2}$. The photon bunching obtained at low temperatures turns to antibunching as thermal mixing between bright and dark exciton states operates. **b** Simulations of the PL autocorrelation function with a four-level model described in Supplementary Note 2, taking $\Gamma_B = 0.9$ ns$^{-1}$, $\Gamma_D = 0.01$ ns$^{-1}$, $\gamma_0 = 0.1$ ns$^{-1}$, $\alpha = 3/4$, $W = 0.01$ ns$^{-1}$, $\eta_{XX} = 0.13$, $E_1 = 3$ meV, $E_2 = 5.6$ meV. **c** Photon coincidence histograms at various magnetic fields, at 4 K, showing the weakening of the bunching effect with increasing magnetic fields. **d** Photon coincidence histograms measured for the same NC in different conditions of line filtering, at 4 K and in zero field. Each histogram is surmounted by the corresponding filtering scheme. Left panel: Histogram recorded in a cross-correlation configuration, selecting the exciton ZPLs on one channel of the correlation setup and the biexciton ZPLs on the other. The asymmetry of the histogram is a signature of ordered biexciton–exciton–zero exciton photon cascades. Right panel: Histogram recorded with high-pass filtering of the exciton ZPLs, showing photon antibunching.

passed through a clean-up filter. The emitted photons are filtered from the scattered laser light with a band-pass filter (centered at 700 nm with a band width of 70 nm) and sent to a single-photon counting avalanche photodiode and a spectrograph (with a resolution of ~120 μeV using a grating of 1800 lines/mm). For resonant PL excitation spectra, a single-mode tunable Ti:Sa laser is used, together with a long-pass filter (with a nominal edge wavelength of 715 nm) whose transmission curve is tunable under a tilt of the incidence angle. PL decays are recorded with a conventional time-correlated single-photon-counting setup using a pulsed laser source (optical parametric oscillator at 561 nm, 200 fs pulse width, with a repetition rate reduced to 8 MHz with a pulse-picker). A Hanbury Brown and Twiss setup is used to build the start–stop coincidence histograms. Filtering of the exciton or biexciton ZPLs is performed with tunable long-pass (with a nominal edge wavelength of 715 nm) and short-pass (with a nominal edge wavelength of 720 nm) filters.

## Data availability
All relevant data that support our experimental findings are available from the corresponding author upon reasonable request.

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

## Acknowledgements

We acknowledge the financial support from the French National Agency for Research, Région Aquitaine, Idex Bordeaux (LAPHIA Program), and the EUR Light S&T. J.E. and B.L. acknowledge the Institut universitaire de France. M.V.K. and A.S. acknowledge financial support from EU via Horizon 2020 [EMPAPOSTDOCS-II program, which received funding under Marie Skłodowska-Curie grant agreement number 754364 and ERC Consolidator Grant SCALE-HALO, grant agreement 819740]. We thank Ihor Cherniukh for electron microscopy imaging.

## Author contributions

A.S., M.I.B., and M.V.K. prepared the samples and performed the ensemble characterization. P.T. and L.H. performed the optical experiments. P.T., L.H., J.-B.T., and B.L. analyzed and interpreted the data. L.B. and Y.L. performed the experiments on CdSe NCs. J.E. developed the model of exchange interaction for the exciton fine structure. P.T., L.H., J.E. and B.L. wrote the manuscript with inputs from all authors. B.L. supervised the project.

## Competing interests

The authors declare no competing interests.
