## [Peer Review File · Nature Communications]

REVIEWER COMMENTS

Reviewer #1 (Remarks to the Author):

The advancements of semiconductor perovskite nanocrystals (NCs) in classical optoelectronic devices and quantum information technologies rely heavily on a thorough understanding of their exciton energy-level structures and photon statistics, which are both addressed in detail in this current report with a combination of theoretical and experimental investigations. Following their previous work on the organic-inorganic hybrid perovskite NCs of FAPbBr₃, the authors move one step further here to show that the lowest-energy state in the all-inorganic perovskite NCs of CsPbI₃ is contributed by the dark excitons. They additionally show that, by increasing the sample temperature and applying the magnetic field, the transition from bunching to antibunching photon statistics can be triggered in a single CsPbI₃ NC, benefiting from the exciton shelving role played by the long-lived dark-exciton state. It is very likely that the above findings, especially the existence of a lowest-energy dark-exciton state, could be extended to perovskite NCs with other compositions and thus, they are well deserved to be published in *Nature Communications*. The authors have successfully addressed most of my previous comments/questions in the last round of review, and I only suggest the following minor changes for them to further improve the manuscript.

In the last sentence of page 3, the authors state that “These spectral structures are assigned to the bright triplet of NCs with cubic, tetragonal and orthorhombic symmetries of the NC shape and/or crystal structure”, which is hard for readers to understand. In addition to the crystal structure, does the shape anisotropy also determine how many PL lines are observed from a single NC? Or does it only affect the order of bright-exciton states and their fine-structure splittings?

Still on this page, in lines 7-8, the authors state “their emission energy centered on 1.77 eV (Fig. 1a) is blue-shifted with respect to the room temperature value”. This is obviously wrong, since they observed red-shifted PL peaks from single CsPbI₃ NCs at the cryogenic temperature.

In their experiment, the single CsPbI₃ NCs are embedded inside a polystyrene matrix and the authors adopt two extreme conditions to simulate the influence of dielectric confinement on the exciton fine structures. I am curious whether they can study the optical properties of single CsPbI₃ NCs without the wrapping polymers, in which case it should be easy to test the correctness of this dielectric confinement model.

The authors mention several times such terms as “a tetragonal NC shape” in the manuscript, which seems to be quite misleading. A single perovskite NC can have a cuboid shape while adopting an underlying tetragonal structure. Then what does “a tetragonal NC shape” exactly mean?

In lines 9-10 of page 7, the authors state that “a non-degenerate sublevel (Z state) defined by the NC elongation axis”. Are there any other factors that should be considered for the definition of the Z state? For example, do we still have a Z exciton if the elongation axis is along the [100] direction of a single CsPbI₃ NC?

In lines 8-9 of page 9, the authors state that “a coherence lifetime $T_2 \sim 10$ ps for the trion emission has been extracted from the first-order PL correlation measurements⁴²”. In ref. 42, the 10 ps was in fact obtained from the quantum interference measurement on the trion absorption

state. Meanwhile, the first-order PL correlation measurement yielded a similar PL linewidth of neutral single excitons that is comparable to the one reported here by the authors from the resonant PLE measurement.

Reviewer #2 (Remarks to the Author):

The revised manuscript is well improved and is acceptable for publication in Nature Communications.

Reviewer #3 (Remarks to the Author):

Report on the revised paper entitled « The dark exciton ground state promotes photon-pair emission in individual perovskite nanocrystals » by Tamarat *et al.*

First, I would like to thank the authors for addressing all the formulated questions even those that I formulated in a very confusing manner. I apologize for the confusing text concerning bunching and antibunching. It was the result of unsightly edit effect in my report (cut of several sentences between text corresponding to the bunching phenomena and text corresponding to antibunching). Finally, concerning in particular this point, my concerns were addressed by the authors since similar and complementary comments were provided by referee 1.

Concerning the fine structure results.

The analysis of the experimental data that authors proposed is now more clear in the manuscript. I agree with them that the Bright-Dark (BD) splitting is not very sensitive to the crystal structure of the nanocrystals and then consider a cubic structure simplify the calculations and it is good choix. Moreover, I underline that the BD splitting is also very slightly depedent on the anisotropy of the considered nanocrystals (see the new Figure 3 and Figures 7 a) and 7 b) in the supplementary information (SI) document).

I agree also with authors that the Bright-Brigt (BB) splittings are very sensitive to the shape anisotropy of nanocrystals and that the simple model that they used (O_h crystal phase + sphape anisotropy) can explain in a very easy way how the inversion of Z state depends on the z elongation of the nanocrystal shape. However this simple model fail to a quantitative description of data represented in Figure 1 a). As we can see in Figure 7a) and 7 b) in SI the BB splitting is clearly underestimated. Authors should add a comment on that.

Concerning the observation of the inversion of Z state, the situation is clear for doublets when a magnetic field is applied as in Figures 6 c), d), e) and f) of the SI document concerning results obtained in $CsPbBr_3$ nanocrystals. However, Figure 2 c) and d) of the SI document for $CsPbI_3$ are consistent with a Z position always placed at low energy and then consistent to the simple model of authors (O_h crystal phase + anisotropy with $\beta > 1$).

However, to identify the order of the levels Z and X or Y in a triplet by arguments taking into account their respective energy positions is only possible if nanocrystals have a O_h (cubic) crystal phase (the hypothesis of the authors). For other crystal phases than O_h the effect of the shape anisotropy is more subtil than the case described in Figures 3 c) and d). The splitting energies, in general, are given by the interplay of anisotropy and crystal phase. Authors write: "yet, the low-temperature crystallographic structure of $CsPbI_3$ NCs is unknown", then the cubic structure is a working hypothesis for the authors. Let us consider a nanocrystal with a tetragonal crystal phase (to take a more simple model), the effect of an anisotropy characterized by $\beta > 1$, for exemple $\beta = 1.13$ (as observed by the authors), is to reduce the BB splitting with respect to the value without anisotropy (see reference 38 in this paper). That is the opposite effect than the one predicted by considering a cubic crystal structure. In this case, it will more difficult to

describe the triple PL spectrum by a group of two close states (corresponding to x and y) and a well-separated Z state. It seems to me that it is then necessary to have another way to identify without ambiguity the corresponding levels X, Y and Z and I agree with authors that is out of the scope of this work but they have to clearly state this point in the manuscript.

Authors propose a new Figure 3. May I suggest some modifications of the Figure captions to make completely clear the data analysis used by the authors:

-caption of figure c): please write that the figure shows "the triplet splitting..... for a tetragonal-shaped NC with O_h crystal phase". For other crystal phases than O_h (cubic) the effect of the shape anisotropy is more subtil than the case described in Figures 3 c) and d). Please, write also for c) that "The splittings are presented for three different NC sizes covering the range of the explored NC sizes.

-caption of figure d):could the authors write that the PL spectra correspond to "slightly distorted tetragonal shape" (thus orthogonal shape)?.

- could the authors give parameters used to calculate long and short range electron-hole exchange interaction in compounds included in figure 3 f)?.

In conclusion, authors have improved the manuscript by giving clearly the hypothesis in which the work is based and addressing different comments and questions of all referees. As I have written in my first report the work represents a leading contribution and provide elements of response to the very current controversy on the nature of the lower energy state in perovskite nanocrystals and about the role of biexcitons and exciton dark state on the photon quantum statistics in these materials. However I still think that this paper is essentially in the continuity with recent results :

-the brightness of dark state (the lowest energy state) by applying magnetic field and the explanation of the band-edge exciton fine structure by electron-hole exchange interaction and morphologie of nanocrystals (the same model was used in Nature Materials by the same authors centered on $FaPbBR_3$ Nanocrystals).

- the exciton relaxation dynamics via a two phonons process (Nature Comm by the same authors on $FaPbI_3$)

- the measurement of the coherence (Science by Utzat H. *et al.*, $CsPb_3$ and Nano Letters by Lv. Y. *et al*, $CsPbI_3$)

- the exciton-shelving role played by the dark state observed previously in CdSe nanocrystals by the same authors (Nano Letters by Louyer *et al*).

That is why I propose transferring the manuscript to Nature Photonics or Nature Communications.

Response to Reviewer #1

We thank the Reviewer for positive comments on our work, and consideration that our “findings ... are well deserved to be published in Nature Communications. The authors have successfully addressed most of my previous comments/questions in the last round of review, and I only suggest the following minor changes for them to further improve the manuscript”.

We develop below the answers to all questions and comments of the Reviewer.

1. In the last sentence of page 3, the authors state that “These spectral structures are assigned to the bright triplet of NCs with cubic, tetragonal and orthorhombic symmetries of the NC shape and/or crystal structure”, which is hard for readers to understand. In addition to the crystal structure, does the shape anisotropy also determine how many PL lines are observed from a single NC? Or does it only affect the order of bright-exciton states and their fine-structure splittings?

We thank the Reviewer for this remark, which helps improving the clarity of our manuscript. Indeed, the bright triplet splittings result from a subtle interplay of crystal structure and shape anisotropy effects. For instance, assuming a cubic crystal structure (resp. shape), shape (resp. crystal structure) anisotropy governs the number of bright levels: Single-line, two-line and three-line spectra will be respectively assigned to cubic, tetragonal and orthorhombic anisotropy of the NC shape (resp. crystal structure).

We have modified the sentence pointed by the Reviewer as follows “..., *these spectral features result from a subtle interplay of crystal structure and shape anisotropy effects^{15,29,30}. For instance, assuming a cubic symmetry of the NC crystal structure (resp. shape), shape (resp. crystal structure) anisotropy governs the number of bright levels : Single-line, two-line and three-line spectra will be respectively assigned to cubic, tetragonal and orthorhombic anisotropy of the NC shape (resp. crystal structure).*”

2. Still on this page, in lines 7-8, the authors state “their emission energy centered on 1.77 eV (Fig. 1a) is blue-shifted with respect to the room temperature value”. This is obviously wrong, since they observed red-shifted PL peaks from single CsPbI₃ NCs at the cryogenic temperature.

We apologize for this mistake, we meant red-shifted. This is now corrected in the manuscript.

In their experiment, the single CsPbI₃ NCs are embedded inside a polystyrene matrix and the authors adopt two extreme conditions to simulate the influence of dielectric confinement on the exciton fine structures. I am curious whether they can study the optical properties of single CsPbI₃ NCs without the wrapping polymers, in which case it should be easy to test the correctness of this dielectric confinement model.

We thank the Reviewer for this interesting suggestion of further spectroscopic investigations of single perovskite NCs without polymer matrix. Yet, in this configuration the NCs would still be in contact with the substrate, which can affect their dielectric confinement. Such studies are very demanding and far beyond the scope of this work.

3. The authors mention several times such terms as “a tetragonal NC shape” in the manuscript, which seems to be quite misleading. A single perovskite NC can have a cuboid shape while adopting an underlying tetragonal structure. Then what does “a tetragonal NC shape” exactly mean?

A tetragonal NC shape means that the shape of the NC has the shape of a crystalline cell in a tetragonal crystal structure (i.e. a cube elongated along one of its axes, with four rectangular faces and two square ones), regardless of its crystal structure.

We have clarified this term in the caption of Figure 3, where NCs morphologies are sketched. We have added “*The cube is elongated along the Z axis, with an aspect ratio defined as $L_x = L_y = L_z/\beta$* ”.

4. In lines 9-10 of page 7, the authors state that “a non-degenerate sublevel (Z state) defined by the NC elongation axis”. Are there any other factors that should be considered for the definition of the Z state? For example, do we still have a Z exciton if the elongation axis is along the [100] direction of a single CsPbI₃ NC?

There is no other factor, the Z axis is defined as the elongation axis of the NC. It may coincide with a symmetry axis of the NC crystal structure, such as the [100] direction.

5. In lines 8-9 of page 9, the authors state that “a coherence lifetime $T_2 \sim 10$ ps for the trion emission has been extracted from the first-order PL correlation measurements⁴²”. In ref. 42, the 10 ps was in fact obtained from the quantum interference measurement on the trion absorption state. Meanwhile, the first-order PL correlation measurement yielded a similar PL linewidth of neutral single excitons that is comparable to the one reported here by the authors from the resonant PLE measurement.

We thank the Reviewer for bringing this point to our attention. In Ref. 42, a coherence lifetime of 75 ps is indeed deduced from first-order correlation measurements on single CsPbI₃ NCs. We have thus replaced the sentence “*For CsPbI₃ NCs, a coherence lifetime $T_2 \sim 10$ ps for the trion emission has been extracted from first-order PL correlation measurements⁴²*” by “*For CsPbI₃ NCs, a similar coherence lifetime has been extracted from first-order PL correlation measurements⁴²*”.

Response to Reviewer #2

We thank the Reviewer for considering that “the revised manuscript is well improved and is acceptable for publication in Nature Communications”.

Response to Reviewer #3

We thank the Reviewer for the positive comment on the fact that authors have “addressed all the formulated questions, even those formulated in a very confusing manner”, and that “improved the manuscript by giving clearly the hypothesis in which the work is based and addressing different comments and questions of all referees”. The Reviewer states that “The analysis of the experimental data that authors proposed is now more clear in the manuscript” and that “the work represents a leading contribution and provide elements of response to the very current controversy on the nature of the lower energy state in perovskite nanocrystals and about the role of biexcitons and exciton dark state on the photon quantum statistics in these materials”.

We develop below the answers to all comments of the Reviewer.

1. “I agree also with authors that the Bright-Bright (BB) splittings are very sensitive to the shape anisotropy of nanocrystals and that the simple model that they used (Oh crystal phase + shape anisotropy) can explain in a very easy way how the inversion of Z state depends on the z elongation of the nanocrystal shape. However this simple model fail to a quantitative description of data represented in Figure 1 a). As we can see in Figure 7a) and 7 b) in SI the BB splitting is clearly underestimated. Authors should add a comment on that.”

We guess that the Reviewer means Figure 3a) instead of Figure 1 a).

Only the contribution of long-range exchange interaction has been taken into account in the simulations of Supplementary Figs. 7a,b. To take into account the contribution of the short-range interaction, a correction factor has to be applied to these splittings, as indicated in the Supplementary Note 1, section 2b. This is done in Fig 3. Additionally, the effects of dielectric confinement, which enhance the calculated fine structure splittings, are not taken into account in Supplementary Fig. 7 and Fig 3c.

In order to clarify this point, we have changed the title of Supplementary Fig. 7: “Band-edge exciton energy *and contribution of the long-range exchange interaction to the fine structure splittings in CsPbI₃ NCs*” and added in its caption that “*The contributions of short-range exchange interaction and dielectric confinement (see Supplementary Note 1) are not taken into account.*”

We have also clarified that Fig. 3 takes into account the short-range exchange interaction, by adding “*which is done in Fig. 3*” in the Supplementary Note 1, section 2b (page 17). We have also added in the caption of Fig. 3c that no dielectric confinement effect is taken into account: “*The effects of dielectric confinement, which enhance the fine structure splittings, are not taken into account in c.*”

2. “Concerning the observation of the inversion of Z state, the situation is clear for doublets when a magnetic field is applied as in Figures 6 c), d), e) and f) of the SI document concerning results obtained in CsPbBr₃ nanocrystals. However, Figure 2 c) and d) of the SI document for CsPbI₃ are consistent with a Z position always placed at low energy and then consistent to the simple model of authors (Oh crystal phase + anisotropy with $\beta > 1$). However, to identify the order of the levels Z and X or Y in a triplet by arguments taking into account their respective energy positions is only possible if nanocrystals have a Oh (cubic) crystal phase (the hypothesis of the authors). For other crystal phases than Oh the effect of the shape anisotropy is more subtil than

the case described in Figures 3 c) and d). The splitting energies, in general, are given by the interplay of anisotropy and crystal phase. Authors write: “yet, the low-temperature crystallographic structure of CsPbI₃ NCs is unknown”, then the cubic structure is a working hypothesis for the authors. Let us consider a nanocrystal with a tetragonal crystal phase (to take a more simple model), the effect of an anisotropy characterized by $\beta > 1$, for example $\beta = 1.13$ (as observed by the authors), is to reduce the BB splitting with respect to the value without anisotropy (see reference 38 in this paper). That is the opposite effect than the one predicted by considering a cubic crystal structure. In this case, it will be more difficult to describe the triple PL spectrum by a group of two close states (corresponding to x and y) and a well-separated Z state. It seems to me that it is then necessary to have another way to identify without ambiguity the corresponding levels X, Y and Z and I agree with authors that is out of the scope of this work but they have to clearly state this point in the manuscript.”

We agree with the Reviewer that the fine structure splittings result from a subtle interplay of shape and lattice anisotropies. Yet, building a model that quantitatively reproduces the spectroscopic findings and rigorously assigns the bright triplet ZPLs to the levels X, Y and Z would require the knowledge, for each NC, of its orientation, 3D morphology, low-temperature crystal structure, as well as dielectric environment. This task is difficult and far beyond the scope of this manuscript.

On page 7, we have added: “More quantitative comparison between theory and experiment, *in particular the assignment of non-degenerate bright triplet ZPLs to the levels X, Y and Z, would require refined models that take into account crystal phase and shape anisotropies*³⁹ together with the knowledge of the low-temperature crystal structure, 3D shape, *orientation and dielectric environment of each NC.*”

3. “Authors propose a new Figure 3. May I suggest some modifications of the Figure captions to make completely clear the data analysis used by the authors:

-caption of figure c): please write that the figure shows “the triplet splitting..... for a tetragonal-shaped NC with Oh crystal phase”. For other crystal phases than Oh (cubic) the effect of the shape anisotropy is more subtle than the case described in Figures 3 c) and d). Please, write also for c) that “The splittings are presented for three different NC sizes covering the range of the explored NC sizes.

-caption of figure d): could the authors write that the PL spectra correspond to “slightly distorted tetragonal shape” (thus orthogonal shape)?.

We have added all this information in the caption of Fig. 3.

- could the authors give parameters used to calculate long and short range electron-hole exchange interaction in compounds included in figure 3 f)?”

The method and parameters used to calculate the short-range exchange interaction of all these compounds had already been given in the Supplementary Note 1. We have added the following sentence in the Supplementary Note 1, page 17: “*In Fig. 3f, the long-range exchange interaction of CsPbBr₃ is also deduced from Ref. ²¹, while for MA- and FA-based hybrid perovskites, the material parameters are obtained from Ref. [Galkowski et al. *Energy & Environmental Science* **9**, 962–970 (2016)]*”. We have added this reference to the list of references of the SI.

List of changes

The corrections to the text are marked in italics.

I- Main text

Page 3, line 9: blue-shifted is replaced by *red-shifted*.

Page 3, the last sentence is replaced by : “*Within a resolution of $\sim 120 \mu\text{eV}$, these spectral structures result from a subtle interplay of crystal structure and shape anisotropy effects^{15,29,30}. For instance, assuming a cubic symmetry of the NC crystal structure (resp. shape), shape (resp. crystal structure) anisotropy governs the number of bright levels : Single-line, two-line and three-line spectra will be respectively assigned to cubic, tetragonal and orthorhombic anisotropy of the NC shape (resp. crystal structure).*”

Page 7, we have developed a sentence: “*More quantitative comparison between theory and experiment, in particular the assignment of non-degenerate bright triplet ZPLs to the levels X, Y and Z, would require refined models that take into account crystal phase and shape anisotropies³⁹ together with the knowledge of the low-temperature crystal structure, 3D shape, orientation and dielectric environment of each NC.*”

Page 8, in the caption of Fig.3 c and d, we have added the following information: “*Computed evolution of the triplet splitting with the aspect ratio β of a tetragonal-shaped NC with a cubic crystal structure. The cube is elongated along the Z axis, with an aspect ratio is defined as $L_x = L_y = L_z/\beta$. (...) The splittings are presented for three different NC sizes (covering the range of the explored NC sizes) and increase with quantum confinement. The effects of dielectric confinement, which enhance the fine structure splittings, are not taken into account in c. d.* Three PL spectra are chosen to illustrate the assignment of the triplet spectral structure to the NC morphology (*slightly distorted tetragonal shape*), assuming a cubic lattice.”

Page 9, we have changed the sentence “*For CsPbI₃ NCs, a coherence lifetime $T_2 \sim 10$ ps for the trion emission has been extracted from first-order PL correlation measurements⁴²*” by “*For CsPbI₃ NCs, a similar coherence lifetime has been extracted from first-order PL correlation measurements⁴²*”.

We have also complied with the editorial requests, by splitting the text into sections.

II- Supplementary Information

Page 8, we have clarified the title of Supplementary 7: “*Band-edge exciton energy and contribution of the long-range exchange interaction to the fine structure splittings in CsPbI₃ NCs*”. In its caption, we have added: “*The contributions of short-range exchange interaction and dielectric confinement (see Supplementary Note 1) are not taken into account.*”

Page 17, (Supplementary Note 1, section 2a) we have added the sentence “*For MA- and FA-based hybrid perovskites, material parameters are obtained from Ref. ²²*”. This reference to *Galkowski et al. Energy & Environmental Science* 9, 962–970 (2016), is added to the list of references.

Page 17, (Supplementary Note 1, section 2b), concerning the contribution of the short-range-exchange interaction, we have added “*which is done in Fig. 3*”.

REVIEWERS' COMMENTS

Reviewer #1 (Remarks to the Author):

The authors have satisfactorily addressed all my technical comments and, with the corresponding revisions, the manuscript is now suitable for being published in Nature Communications.

Reviewer #3 (Remarks to the Author):

Report on the revised paper entitled "The dark exciton ground state promotes photon-pair emission in individual perovskite nanocrystals" by P. Tamarat et al.

The authors have considerably improved the manuscript and taken into account comments addressed by referees. There is only some small points that have to be modified before publication, but if authors approve these changes a feedback is not really needed and I will consider that the manuscript will be ready for publication in Nature Communications.

- The requested modifications concern the absence of SR contribution to the fine structure bright-bright splitting for nanocrystals with cubic crystal phase. (See formula 55 in reference [38] in main text of this manuscript).

-According with this fact, the caption of figure 7 in SI should be changed: Instead to write "...The contributions of short-range exchange interaction and dielectric confinement (see Supplementary Note 1) are not taken into account..." authors should write "...The contribution of dielectric confinement (see Supplementary Note1) is not taken into account...'

- In SI Note 1b) the following sentences should also be modified:

- Instead write: "The short-range contribution of the exchange interaction is added to the long-range contribution to compute the fine structure splittings". Authors should write : "The short-range contribution of the exchange interaction is added to the long-range contribution to compute the bright-dark splitting"

- Instead write: "Taking into account the SR contribution in the fine structure splittings of CsPbI₃ NCs results in multiplying the splittings calculated with the LR contribution alone by a factor of 1.227, which is done in Fig. 3." Authors should write: "Taking into account the SR contribution in the bright-dark splitting of cubic CsPbI₃ NCs results in multiplying the splitting calculated with the LR contribution alone by a factor of 1.227, which is done in Fig. 3b